# Effect of Acidic Electrolysed Water and Pulsed Light Technology on the Sensory, Morphology and Bioactive Compounds of Pennywort (*Centella asiatica* L.) Leaves

**DOI:** 10.3390/molecules28010311

**Published:** 2022-12-30

**Authors:** Siti-Zaharah Rosli, Noranizan Mohd Adzahan, Roselina Karim, Nor-Khaizura Mahmud Ab Rashid

**Affiliations:** 1Department of Food Technology, Faculty of Food Science and Technology, Universiti Putra Malaysia (UPM), Serdang 43400, Selangor, Malaysia; ara.ochie@gmail.com (S.-Z.R.); rosaz@upm.edu.my (R.K.); 2Department of Healthcare and Professional, Faculty of Health & Life Sciences, Management & Science University (MSU), University Drive, Off Persiaran Olahraga, Section 13, Shah Alam 40100, Selangor, Malaysia; 3Department of Food Science, Faculty of Food Science and Technology, Universiti Putra Malaysia (UPM), Serdang 43400, Selangor, Malaysia; norkhaizura@upm.edu.my; 4Laboratory of Food Safety and Food Integrity, Institute of Tropical Agriculture and Food Security, Universiti Putra Malaysia (UPM), Serdang 43400, Selangor, Malaysia

**Keywords:** pulsed light, electrolysed water, *Centella asiatica*, ‘ulam’, triterpene, minimal processing, leafy greens, sensory

## Abstract

Pennywort (*Centella asiatica*) is a herbaceous vegetable that is usually served in the form of fresh-cut vegetables and consumed raw. Fresh-cut vegetables are in high demand as they offer convenience, have fresh-like quality and are potentially great for therapeutic applications. However, it could be the cause of foodborne outbreaks. Pulsed light is known as a decontamination method for minimally processed products. The aim of this study was to determine the influence of pulsed light in combination with acidic electrolysed water on the sensory, morphological changes and bioactive components in the leaves of pennywort during storage. A combination of soaking with acidic electrolysed water (AEW) at pH 2.5 and pulsed light (PL) treatment (1.5 J/cm^2^) was tested on the leaves of pennywort. After treatment, these leaves were refrigerated (4 ± 1 °C) for two weeks and evaluated on the basis of sensory acceptance, the visual appearance of the epidermal cell and bioactive compounds. In terms of sensorial properties, samples treated with the combined treatment were preferred over untreated samples. The combination of AEW and PL 1.5 J/cm^2^ was the most preferred in terms of purchasing and consumption criteria. Observations of the epidermal cells illustrated that PL treatment kept the cell structure intact. The bioactive phytocompounds found in the leaves of pennywort are mainly from the triterpene glycosides (asiaticoside, madecassoside, asiatic acid and madecassic acid) and are efficiently preserved by the combined treatment applied. In conclusion, the combination of acidic electrolysed water and pulsed light treatment is beneficial in retaining the sensory quality and bioactive compounds in the leaves of Pennywort during storage at 4 ± 1 °C.

## 1. Introduction

Over the last few decades, minimally processed fruits and vegetables have grown in popularity as important components of the human diet [1]. This increase in consumption has sparked an interest in the nutritional, sensory and microbiological quality of these products [2]. With the consumer’s growing awareness of the benefits of balanced diets, minimally processed vegetables have become a preference due to their nutritional value and convenience [3]. Market research shows that 26 percent (%) of people buy pre-cut fruits and vegetables because they are easier to use and 20 percent of people like to see more ready-to-eat single-serving fruits and vegetable options on the market [4].

Pennywort is normally consumed in a raw form among South East Asian populations such as in Indonesia, Thailand and Malaysia [5]. Pennywort or ‘pegaga’ in the Malaysian local dialect, is a fragrant weekly herb of the family Umbelliferae [6]. It is a trailing herb that creeps and is perennial and slender. It prefers moist or wet soils and full sun but tolerates partial shade. Besides pennywort or ‘pegaga’ (*Centella asiatica*), ‘petai’ (*Parkia speciosa*), ‘selom’ (*Oenanthe javanica*) and ‘ulam raja’ (*Cosmos caudatus*) are the most often consumed ‘ulam’ by Malaysians [5]. ‘Ulam’ is a generic Malay word for the freshly eaten plant, and the closest English translation for the word ‘ulam’ is salad [7]. For hundreds of years, pennywort has also been widely used in folk medicine to cure a wide variety of ailments [8]. In India, pennywort is known as ‘mandukaparni’ and is utilised in Ayurvedic treatment. Meanwhile, pennywort is classified as a Traditional Chinese Medicine (TCM) in China [9]. It has been used to aid in the healing of minor wounds. Apart from wound healing, the plant is used for the treatment of a variety of skin ailments, including lupus, leprosy, varicose ulcers, psoriasis, eczema, diarrhoea, amenorrhea, fever and diseases of the female genitourinary system, as well as anxiety relief and cognitive enhancement [10].

Pennywort leaves are abundant in bioactive components that possess a wide range of biological activities and therapeutic values [11]. Several main components present in the leaves of pennywort, such as triterpenes including madecassoside, madecassic acid, asiaticoside and asiatic acid, have been widely researched for therapeutic applications, including ulcers and inflammatory illnesses [9]. Asiaticoside (molecular formula: C_48_H_78_O_19_), a major active constituent of pennywort, plays an important role in healing gastric ulcers [12]. Triterpenes have unique biological and pharmacological properties and are found in all species, mainly plants [13]. Natural triterpene compounds have varied physiological effects. For example, triterpenes show cytotoxic properties against tumour cells with low activity towards normal cells [14]. Triterpenes have been shown to possess anti-viral, anti-bacterial, anti-fungal, anti-inflammatory and anti-oxidative properties, as well as anti-cancer and chemopreventive. They inhibit neoplastic cell lines and induce cancer cell apoptosis, causing their ‘suicidal’ death without harming normal cells [15].

Nevertheless, studies have highlighted the potential of soil and poultry manure as sources of contamination of raw vegetables [16]. Public health has been seriously affected by the sharp increase in outbreak cases linked to the consumption of fresh vegetables [17]. These outbreaks are caused by several pathogenic bacteria, such as *Salmonella*, *Escherichia coli* and *Listeria monocytogenes*, which are linked with fresh-cut products or raw vegetables [18]. *Aspergillus flavus* found in soil, plants and other substrates has also been reported as a pathogen for plants and a carcinogen-producing fungus for humans [19]. Several nonthermal treatments, including as intense pulsed light (IPL), radiation, pulsed electric field, ultrasound and ozonated water, have been developed for commercial use to kill microbes, retain nutritional value and increase the shelf life of fresh-cut items [20,21].

Nonthermal technology is a sustainable technique in line with the sustainable development goals and Agenda 2030 issued by the United Nations (UN) and can be applied in the food industry for the purposes of reducing energy consumption, lowering energy costs and providing assistance to society by reducing the use of energy resources and the emission of numerous air pollutants such as carbon dioxide [22]. Moreover, technologies such as pulsed light (PL), ultraviolet light (UV-C), gamma radiation, ultrasound and electrolysed water have been tested for commercial use to kill bacteria, preserve nutrients and extend the shelf life of fresh-cut items [20,21,23,24]. Compared to conventional thermal processing, nonthermal solutions can help preserve the quality of fruit and vegetables more effectively [25]. It is the potential to sterilise, preserve, or improve the nutritional and sensory qualities of food for its preservation by using a potential nonthermal technology, namely the pulsed light (PL) treatment [26]. Numerous studies have revealed the potential application of PL treatments to inactivate pathogens and reduce spoilage microorganisms under conditions that have minimum impact on sensory or nutritional characteristics in order to increase shelf life [27,28,29,30]. It has been demonstrated that pulsed light effectively reduces bacteria counts on food contact materials, the surfaces of products and medical equipment [31].

The term ‘hurdle technology’ or ‘combination technique’ refers to the sequential or simultaneous use of two or more food preservation techniques to improve food safety and quality while reducing the intensity of certain treatments [32] and to improve food’s microbiological stability, sensory quality, nutritional value and economic properties [33]. The limits of various microbe elimination methods have caused the development and acceptance of hurdle technology to enhance the safety, nutritional value and sensory attributes of goods [34].

In the fresh-cut fruit industry, quality is largely determined by appearance which includes surface colour, the presence or absence of discoloration, desiccation or shrinking, visible microbial growth and the presence or absence of any other defects [35]. The sensory appeal is also the primary factor impacting customers’ purchase intention, consumption and satisfaction with convenience foods, including fresh-cut fruits [36]. In accordance with that, many authors have summarised the advantages of electrolysed water over chlorine [37,38] such as on-site and simple production, cheap and easy-to-find raw materials (water and NaCl), low operational expenses and lower trihalomethane generation. The use of electrolysed water, either acidic, alkaline, or a combination, is able to reduce microbes while maintaining the physicochemical and sensory properties of the leaves of pennywort [39].

Considering the potential of the leaves of pennywort, its main compounds (madecassoside, madecassic acid, asiaticoside and asiatic acid) were chosen as the primary focus of this investigation. Thus, the aims of this study were to determine the influence of pulsed light in combination with acidic electrolysed water on the sensory, morphological changes of the leaves of pennywort, and bioactive components in the leaves of pennywort.

## 2. Results and Discussion

### 2.1. Sensory Acceptance Quality

The sensory quality of fresh produce is an important attribute required by consumers. Attributes such as freshness, brightness, colour, absence of defects and cleanliness are essential for product acceptance and influence the sensory shelf life of vegetable products [40]. The results of the five sensory attributes of the acceptability of appearances, namely colour, freshness, physical damage, brightness and overall quality that were examined by the untrained panels are illustrated in Figure 1a–d.

The mean values for the overall acceptance attribute are shown in Figure 1e. From the findings, it can be concluded that sensory attributes of leafy vegetables deteriorate with time after harvest. Processing procedures can hasten the degradation of the product due to cutting, peeling, cleaning and packing processes, by causing biochemical, sensory and microbiological changes on the plant tissue surface [41]. However, the results of using a combination technique gained more acceptability preference from consumers compared to untreated samples. This finding support other analyses where the quality of safety, texture and colour improve with combination techniques. The cut-off points of a score of three (3) was used to indicate the acceptability of the panellist on the leaves of pennywort as a result of using a combination technique by using hedonic score based on 5-point scales. This is correlated to the limit of acceptance denoted by score three (3) and above [42]. Hence, the dominant score at treatment AEW with PL 1.5 J/cm^2^ was the most accepted treatment for sensory acceptance quality.

Figure 2 shows the sensory evaluation score on yes percentages (%) of fresh-cut leaves of pennywort. 

The survey question, consisting of questions 1 (Q1) and 2 (Q2), elicited the percentages on the acceptability evaluation of fresh-cut leaves of pennywort during storage. The sensory score on the yes percentage was presented with Q1 questions on buying consideration criteria and Q2 questions on consuming criteria. In general, the yes percentages reduced as the storage day increased. The control sample showed a reduction in buying consideration criteria from 88% on day 0 to 20% on day 8 (Figure 2a). In addition, the percentage of panellists who would purchase the product if it were available for AEW + PL1.5 from day 0 to day 12 decreased from 95% to 34%. A total of 34% of the panellists said they would buy the product if it were on the market on day 12 for AEW + PL1.5. However, the remaining 66% said they were reluctant to buy products on the market because of the appearance of the produce.

On the other hand, in terms of consuming criteria in Q2, the results indicated a higher percentage (41–47%) of consuming a product if the product was available in the refrigerator on day 12 for AEW and AEW + PL1.5. The results also indicated that sensory evaluation appearance and consumers ‘yes and no’ survey play important roles in consumers’ preference and acceptability of pennywort. This study contributes to the literature on the acceptability and preference of pennywort and provides practical implications to help farmers and food retailers in the ready-to-eat vegetable market.

### 2.2. Scanning Electron Microscopic (SEM)

The DNA of a plant stores information on how its leaves are structured [43]. Leaves are one of the vegetative organs of vascular plants. Its function synthesises organic matter through photosynthesis and provides the root system with the power to absorb water and mineral nutrients through transpiration. However, each plant has a distinctive leaf shape. Even members of the same family can have drastically diverse leaf forms.

The techniques for visualising the surface, along with the stomata, pore and other parts of the leaves, offer insights into the inner membrane and its effect on the cell organelles, utilising SEM and TEM as a tool, are noticed. However, scanning electron microscopy (SEM) is often used to study the surface morphology and physiological state of plant leaves. Using the image from a scanning electron microscope (SEM), the structure of the leaves of pennywort was observed to examine the influence of acidic electrolysed water and PL treatment on the quality of freshly cut leaves of pennywort.

Plants employ transpiration to transport water from the soil to their vascular network, which spreads from their roots to their leaves. When water vapour reaches the stomata (pores), it is expelled into the surroundings [44]. Guard cells are specialised cells that enclose stomata and work to open and close the stomatal pores. Guard cells are large crescent-shaped cells, two of which enclose a stoma and are attached at both ends. These cells expand and contract to allow stomatal openings to open and close. Additionally, guard cells include chloroplasts, the organelles in plants that capture light.

The outline of individual epidermal cells, guard cells, the stomatal opening and the inner cell walls can be easily noticed (Figure 3a,b). Shrinkage of epidermal cells was induced (Figure 3c,d). There was a clear outline of both epidermal cells and guard cells, and both guard cells were easily distinguishable. This shows that the sample started to shrink because it was kept for a long time (Figure 3b,d) for the untreated (control) sample of the leaves of pennywort. Therefore, based on observations of the outline of individual epidermal cells, the size of guard cells does not differ among leaf samples. However, their forms are enlarged and their surfaces are wrinkly.

After storage, there is a degradation of the leaf structures, as evidenced by stomata that have sunk and cellular morphology with decreased turgor, as shown in Table 1. 

Stomata are the pores on a leaf surface controlling gas exchange, mainly CO_2_ and water vapor, between the atmosphere and plants and thus regulate carbon and water cycles in various ecosystems [45]. In the texture analysis, no negative influence on the leaf structure after washing and PL treatment was found. Throughout the storage period, cell shrinkage in the shape of leaf structure was seen in this test, particularly around guard cells and in the centre of epidermal cells. Nonetheless, the data revealed no significant differences between the treated samples. The slight variations were probably due to the presence of leaf structure in the scanned section.

SEM is an effective tool for selecting the best microscopic method for studying leaf surfaces. On day 12, a decrease in intact epidermal cells and partial shrinkage of epidermal cells were noted for all treatments, with a clear observation found for the acidic EW 4.2 and 6.9 J/cm^2^ treatments. In addition, on day 12, acidic EW 1.5 J/cm^2^ showed a better visual observation on epidermis cells as well as guard cells, stomatal pores and inner cell walls compared to other samples.

### 2.3. Transmission Electron Microscopic (TEM)

Transmission electron microscopy (TEM) has been considered to be the most common technique to investigate the ultrathin sections at microlevels [28]. Ultrathin sections obtained from the mesophyll of the leaves of pennywort were observed. From the outline’s images, cell walls, chloroplasts, ribosomes and vacuoles were observed in the leaves of pennywort (Figure 4).

There were numerous intracellular features, such as nuclei, vacuoles, mitochondrion, golgi apparatus, chloroplast, endoplasmic reticulum, ribosomes and lysosomes, which are typical of algal cells. The double-staining technique, cell walls, nucleus, chloroplast, golgi apparatus, mitochondrion, pyrenoid and starch granule were well resolved in the ultra-low lead staining (ULLS) prepared for TEM analysis [46].

Throughout storage, the weakening and damage of cell walls were observed. TEM micrographs also showed that samples with treatment in storage delayed the degradation of pectin and protected the integrity of cell structure. During early storage (day 0), the epidermal cells were dense and thready, with granular material in the vacuoles and intercellular gaps. Degradation was observed on day 12 (see Figure 5). Thus, the intercellular spaces of the epidermal cells were loosened throughout storage for all samples. However, between the control sample and PL treatment at 1.5 J/cm^2^, PL treatment was found to be more effective in retaining the structure intact.

### 2.4. Phytochemical Screening of Centella asiatica

The phytochemical screening of pennywort analysis focuses primarily on chemicals for which a matched library obtained a similarity index of 80% [47]. In the study, GCMS chromatogram detected the presence of fifteen significant compounds in the ethanol extract of the pennywort leaves from control and PL-treated samples (Figure 6, Table 2), reflecting the proportion of composition of the leaves as in crude extract. These primary and secondary metabolites include phenolic, phytosterol, fatty acid and organic acids.

First, the phytochemical screening of the ethanolic extracts showed that leaves of pennywort had terpenoids, flavonoids, fatty acids, steroids, amino acids and protein. The therapeutic effects of pennywort may be linked to the existence of bioactive metabolites. 2,3-Dihydro-3,5-dihydroxy-6-methyl-4H-pyran-4-one (DDMP), indicating its potent antioxidant activity [48]. Previous research has demonstrated that DDMP belongs to the flavonoid group [49]. In addition, there were components of fatty acid (such as linoelaidic acid, linolenate, octadecadienoic acid and many others) and phytosterol/sterol (such as stigmasterol) (listed in Table 2). In a prior investigation, chemical elements such as linoleic and palmitic acid in the fatty acid group and certain sterols are also discovered.

Moreover, pennywort leaves contain a wide range of diverse and complex chemical elements, including terpenes in a variety of chemical classes, including neophytadiene [50]. Pentacyclic triterpenes were among the active chemicals in the leaves of pennywort. Triterpene substances, including sterols and triterpenes, can accumulate in plants as glycosides (saponins) in massive amounts [51]. Pennywort contains both forms of asiatic acid, which are asiaticoside and free triterpene [50]. Triterpenes are one of the most prevalent natural compounds, with about 30,000 different structures having been discovered so far [52]. Triterpenes are a subset of terpenes, one of the most prevalent classes of natural compounds [53]. All terpenes originate from C5 isoprene units, and they are divided into hemiterpenes (C5), monoterpenes (C10), sesquiterpenes (C15), diterpenes (C20), sesterterpenes (C25), triterpenes (C30) and tetraterpenes (C40) depending on the number of isoprene units [54]. The triterpenoid saponins, known as centroids, were the most significant elements identified from the leaves of pennywort [55].

As terpenes are widely employed as quality control makers for pennywort. As a result, utilising UHPLC to quantify the terpenes as madecassoside (MS), asiaticoside (AS), madecassic acid (MA) and asiatic acid (AA) as a standard was necessary.

### 2.5. Quantification of the Triterpenes

The UHPLC analysis was used to simultaneously determine madecassoside (MS), asiaticoside (AS), madecassic acid (MA) and asiatic acid (AA). The measurements were performed to determine the mean amount of MS, MA, AA and AS in each sample. The levels of MS, MA, AA and AS were compared in different treatment and storage days. Table 3 shows the level of the four analytes in the samples.

The levels of MS and AS were found to be the most dominant, whereas the levels of MA and AA were considerably low. This agrees with a previous study [56]. For the MS level, our result indicated the reduction values from day 0 to day 12 for all treatments. While during storage, MA, AA and AS showed decreased values for untreated (tap water) treatment and increased values for acidic EW, acidic EW 1.5 J/cm^2^, acidic EW 4.2 J/cm^2^ and acidic EW 6.9 J/cm^2^ treatments. In addition, PL treatment was able to keep the bioactive level higher than the untreated sample on day 12. PL treatment significantly increased the concentration of volatiles related to green and herbaceous odours [57]. These results show that the levels of chemical components in the leaves of pennywort could be markedly affected by differences in their environmental growth conditions (climate and soil fertility), harvest and post-harvest processing [56].

## 3. Materials and Methods

### 3.1. Materials

Bundles of fresh pennywort at commercial maturity were purchased in sufficient quantity for all the experiments from a commercial farm in Paya Rumput, Melaka. Pennywort leaves with commercial maturity of 60 days were adopted from Rosalizan et al. [58]. From the bundles, only fresh pennywort leaves with regular and uniform shapes and without any defects were selected. Bruised and yellow leaves were discarded and cut using sterilised stainless-steel knives. Plant taxonomic identification was then conducted by a Herbarium Officer from the Institute of Bioscience (IBS), Universiti Putra Malaysia, under the voucher specimen number SK 3004/16.

### 3.2. Preparation of Sample

The selected leaves of fresh pennywort were washed for approximately 2 min under slow-running tap water to remove soil and dirt [59]. The leaves were plucked off from the stems. The stems were removed, but the petioles of approximately 5 cm from the top of the leaves were still used. The plucked leaves and petioles were air-dried at room temperature (25 ± 2 °C) for 20 min and packed into polypropylene bags (thickness of ~40 μm, length of 125 mm and width of 170 mm). The bags were sealed before exposure to PL treatment. The packed samples were kept refrigerated for storage studies.

### 3.3. Washing (Pre-Treatment)

The pennywort sample (0.050 kg/L) was submerged in acidic electrolysed water (AEW) in a beaker for five (5) minutes. The AEW was obtained using an electrolysed water generator (Leveluk SD 501, Enagic Co., Japan). The status of AEW was measured by a pH meter (Mettler-Toledo AG, Switzerland) with a glass electrode. The samples were removed using sterile forceps. After the excess water was allowed to drain, the samples were air-dried at room temperature (25 ± 2 °C) for 20 min. The samples were then packed in five (5) grams and sealed in a sterile polypropylene plastic bag for PL treatment. The samples without PL treatment were put aside and analysed together with PL-treated samples.

### 3.4. Apparatus

The pulsed light treatment was carried out using a pulse light machine (Steribeam XeMaticA-2L Kehl, Germany) that was composed of two lamps, one above and another below a quartz table that was placed at the centre to hold samples with a maximum energy emission of 350 J. The distance between the light and the sample shelf was 10 cm. The emitted spectrum wavelengths ranged from 180 nm to 1100 nm. The total fluence per pulse emitted was 0.3 J/cm^2^.

### 3.5. Pulsed Light Treatment

The samples were treated with pulsed light using an automatic laboratory flash lamp system (Steribeam XeMaticA-2L Kehl, Germany). The samples were exposed to PL for 1, 3 and 5 min at varying numbers of pulses (5, 14 and 23, respectively). The selected fresh-cut pennyworts were treated at three different fluences (1.5 J/cm^2^, 4.2 J/cm^2^ and 6.9 J/cm^2^). The fluence of pulsed light was calculated by using the following formula:Fluence (J/cm^2^) = Number of pulses × total fluence per pulses
where total fluence per pulse = 0.3 J/cm^2^.

Example:Fluence (J/cm^2^) = Number of pulses × total fluence per pulses 
where 1 min = 5 pulses and total fluence per pulse = 0.3 J/cm^2^
Fluence (J/cm^2^) = 5 *×* 0.3 J/cm^2^ = 1.5 J/cm^2^


### 3.6. Sensory Evaluation

There were two (2) parts to the acceptability survey. In the first part, panels were given different packs of pennywort leaves to measure the acceptability of the appearance of the leaves. In the second part of the acceptability survey, consumers responded to a set of ‘yes or no’ survey questions to measure their intention to purchase and consume the samples. Seventy participants participated as panellists. The comprised panellists were between the age of 20 to 45 years. The criteria for participation used were that those who consume the item (i) be at least 18 years of age, (ii) be the primary shopper for the household, (iii) purchase and eat ‘ulam’ regularly and (iv) like to eat fresh-cut produce.

Each consumer was served eight (8) packs of pennywort leaves, corresponding to four (4) storage intervals. The packages were sealed in odourless plastic containers labelled with three random numbered digits. Before the evaluation, the samples were put on a tray at room temperature while the panellists were given a short briefing on the evaluation procedures and the use of the signal lights in the sensory booths. The panellists were seated in individual partitioned booths in a sensory evaluation laboratory.

#### 3.6.1. Acceptability Test

The technique used was affective (consumer) tests using the hedonic acceptance test. The evaluations were repeated in two batches. The panellists were asked to evaluate the colour, freshness, physical damage, brightness and overall quality of pennywort leaves based on a 5-point scale based on the appearance of the leaves without consuming them. The 5-point hedonic category scale, ranging from 1 (the least intense) to 5 (the most intense and acceptable condition), was used to measure the intensity of each sensory attribute for the different treatments of the leaves of pennywort at different storage times. The overall quality was also rated using the same scale.

#### 3.6.2. Survey Question

For the second part of the ‘yes’ and ‘no’ sensory survey questions, consumers evaluated the aesthetics of each sample and responded ‘yes’ or ‘no’ to the queries. The questions were as follows:

**Question** **1.**
*‘Visualise you are in a grocery store. You want to buy a minimally processed pennywort. If you found a package of pennywort with leaves like these, would you usually buy it?’*


**Question** **2.**
*‘Let’s say you have a leaf of pennywort in your fridge. Would you usually eat it?’*


The question order was not expected to have any influence on the results as consumers consume their purchased products. Issues were consistently given in a similar attempt to consumers buy a product, store it in their fridges and finally consume it. As a result, consumers’ responses while visualising the pack of pennywort leaves similar to the one observed were considered indicative of their real choice. The studies were carried out in a sensorial facility that featured separate cubicles with illusionary sunlight brightness, temperature regulation (22–24 °C) and airflow.

### 3.7. Scanning Electron Microscopy (SEM)

The scanning electron microscopy (SEM) analysis was practiced in a biosafety cabinet, and prior to the analysis technique adhered to the instructions according to the standard method for plant samples. Pennywort leaves were cut into three 0.5 to 1 cm slices with a sterile scalpel, put in separate vials and fixed in fixative (4% Glutaraldehyde) for two (2) days at 4 °C. The leaves were washed with 0.1 M sodium cocodylate buffer for 30 min. This step was repeated twice. Samples were then fixed in 1% osmium tetroxide for 2 h at 4 °C. Samples were rinsed three times for 30 min each with 0.1 M Cacodylate Buffer and then dehydrated by acetone gradient series of 35%, 50%, 75%, 95% and 100%.

Next, 100% ethanol was applied twice to ensure a complete dehydration [60]. The exposure in each step was 30–45 min, with the final concentration being repeated 3 times for 1 h. Then, the samples were dried using a critical point dryer (CPD) (Leica EM CPD 030) and followed by mounting onto specimen stub for coating with a thin layer of gold-palladium (Baltec SCD 005). Different locations of the coated samples using a scanning electron microscope (JEOL JSM-IT 100) were used to study the morphology of pennywort leaves.

### 3.8. Transmission Electron Microscopy (TEM)

The leaves of pennywort were sliced into 2–4 mm cubes with a double edge razor blade and fixed for 4 h in 2.5% glutaraldehyde in 100 mM cacodylate buffer (pH 7.2). After fixation, the tissues were rinsed 4 times in distilled water post-fixed for 1 h with 1% aqueous osmium tetroxide. Following four rinses in distilled water, the seed tissue was dehydrated in a graded acetone series and infiltrated with Spurr’s resin.

Resin-embedded sample tissue was polymerised at 65 °C oven for 48 h. Ultrathin sections of the sample tissue were cut with a diamond knife, collected on 200 mesh copper grids and stained with 0.5% uranyl acetate and 0.4% lead citrate. The grids were examined under the JEM-2100F (Tokyo, Japan) transmission electron microscope. The samples of leaf tissues were fixed in a mixture of 3% (*v*/*v*) glutaraldehyde and 2% (*w*/*v*) paraformaldehyde in 0.1 mM phosphate buffer, pH 7, overnight at room temperature.

Samples were subsequently post-fixed with 1% (*w*/*v*) osmium tetroxide in the same buffer for 1 h at 4 °C. They were dehydrated in a graded ethanol series before being embedded in SPURR resin (Electron Microscopy Sciences, Washington, PA, USA). Ultrathin sections were cut from at least three leaf samples from different plants and contrasted with uranyl acetate and lead citrate before being examined with a Jeol JEM-100 SX TEM at 80 kV and Zeiss LIBRA 200FE-HR TEM (transmission electron microscope), operating at 200 kV and equipped with an in-column omega filter for energy selective imaging.

### 3.9. Crude Extracts Preparation

The crude extract preparation was conducted according to Rafi et al. [56] with slight modification. The pennywort leaf samples were dried under the shade for 24 h and powdered with a blender. The dried leaf powder (5 g) was added to 100 mL ethanol (75%) and sonication-assisted extraction was done. After that, the samples were placed at room temperature in seal Scott bottle for 48 h for the maceration. The resulting mixture of ethanol extract was filtered under vacuum using a Buchner funnel with Whatman No 1 filter paper and the residual plant materials were subjected to further maceration with fresh solvent up to three times to ensure exhaustive extraction. Individual ethanol filtrate was combined and concentrated under reduced pressure at 40 °C using the rotary evaporator (Buchi Rotavapor, R-114, Fawil, Switzerland) The crude extract was used for GS-MS and UHPLC analysis examinations.

#### 3.9.1. Sonication-Assisted Solvent Extraction

Sonication-assisted extraction is a modified maceration in which ultrasound is utilised to improve extraction efficiency. The mixture of crude extract was placed in a closed container, similar to the maceration period. The container was then placed in an ultrasonic bath. In such a condition, ultrasound transfers the mechanical power onto the plant cells, leading to the breakdown of cell walls and the increased solubilisation of extracts in the solvent. The sonication time used for the mixture was 1 h with a power of 256 W to assist the extraction before maceration.

#### 3.9.2. Preparation of Samples and Standards

Approximately 100 mg of powdered materials were soaked in 5 mL of methanol. They were ultrasonically processed for 1 h at ambient temperature using an ultrasonicator (Branson 1510E-MT 42 kHz, Danbury, CT, USA) for sample extraction. Before submitting the sample solutions for UHPLC analysis, they were filtered through a membrane with a pore size of 0.22 microns and diluted to a volume of 10 mm using methanol. Madecassoside (87.5%), madecassic acid (88.1%), asiatic acid (92.6%) and asiaticoside (88.8%) were obtained from ChromaDex Inc (Los Angeles, CA, USA). The solvents used in the UHPLC analysis were of analytical HPLC grade (Merck, Darmstadt, Germany). Whatman membrane filters (0.22 μm pore size; PTFE; P/N E252, Buckinghamshire, England) were used for the filtration of sample solutions.

It was found that the primary bioactive components in pennywort leaves were asiatic acid (AA), madecassoside (MS), madecassic acid (MA) and asiaticoside (AS). In methanol, 1000 g/mL solutions of MA, MS, AA and AS, were made to be used as standards. To obtain calibration curves for MS, AS, MA and AA, an adequate volume of each standard solution was mixed and diluted with methanol at six (6) concentrations ranging from 100 to 500 μg/mL.

### 3.10. Gas Chromatography-Mass Spectrometry (GC-MS)

To investigate the chemical fingerprinting of different crude extracts, the GC-MS analysis was performed using the Shimadzu GC-2010 Plus apparatus gas chromatograph was equipped with Shimadzu GCMS-QP2010 Ultra mass spectrometer and used Rxi 5MS steel column (30 mm × 0.25 mm × 0.25 µm). The carrier gas used was helium at constant pressure (37.1 kPa), with the oven temperature programmed from 50 °C and raised at 3 °C/min to a final temperature of 300 °C (hold time 10 min). The injector temperature was 250 °C, with an injection volume of 1 µL. The ion source temperature was 250 °C at the interface of 250 °C. The solvent cut time was 3 min and the total run time was 93 min. A continuous flow rate of 1 mL/min was employed for the carrier gas (helium) before a 1 mL sample was injected. In full scan mode, the mass spectral scan range was adjusted from 40 to 700 (m/z). The chemicals were identified using the National Institute of Standards and Technology (NIST) database. Their retention index was matched to those of legitimate samples deposited on Wiley. This was similar to the strategy employed by [61] with some modifications.

### 3.11. Ultra-High Pressure Liquid Chromatography (UHPLC)

Briefly, the UHPLC analysis was performed using the Ultimate 3000 series UHPLC system (Thermo Fisher Scientific, Waltham, MA, USA), whereas the chromatographic separation was performed using the Thermo Hypersil aQ column (100 mm × 2.1 µm × 1.9 µm) maintained at 40 °C. For testing, a sensor wavelength of 206 nm was used. The detector was a photodiode array.

The flow rate used was 0.4 mL/min, and the injection volume of samples was 5 µL with mobile phases comprising 0.1% formic acid in water (solution A) and 0.1% formic acid in acetonitrile (solution B). A device controller and data collector were involved in the operation with Xcalibur 2.2 software (Thermo Fisher Scientific, Waltham, MA, USA). A linear gradient elution was applied as follows: (1) 5% eluent B for 10 min, (2) a gradient from 5% to 65% eluent B over 30 min, (3) holding for 5 min, (4) returning to 5% eluent B in 35 min and (5) holding for 5 min.

### 3.12. Experimental Design

The leaves of pennywort were packed (5 g) and divided into five (5) groups. These groups were (i) control (tap water), (ii) acidic electrolysed water (AEW) at pH 2.5, (iii) AEW and pulsed light (PL) at fluence 1.5 J/cm^2^, (iv) AEW and pulsed light (PL) at fluence 4.25 J/cm^2^, and AEW and (v) pulsed light (PL) at fluence 6.9 J/cm^2^.

The leaves were immersed in a glass beaker, as described by Siti Zaharah et al. [39], for five (5) minutes before being dried and packed. The leaves were then arranged in polypropylene plastic packaging before being exposed to PL treatment. After treatment, all samples were stored at 4 ± 1 °C.

The aims were to determine the sensory quality on the appearance acceptability test, morphological changes in the leaves of pennywort during storage using scanning electron microscope (SEM) and transmission electron microscopic (TEM) and the bioactive compound in the leaves of pennywort. The leaves were extracted to obtain the crude extract for the bioactive compound screening. The existence of active components was screened using gas chromatography-mass spectrometry (GC-MS) for control and PL treatment at 1.5 J/cm^2^. The components were then quantified using high-pressure liquid chromatography (HPLC) analysis.

### 3.13. Statistical Analysis

The experiments were performed in triplicates. The data were analysed using Minitab version 18 statistical package (Minitab Inc., Minneapolis, MN, USA) based on the analysis of variance (ANOVA) and expressed as mean value ± standard deviation. The confidence level for statistical significance was set at a probability value of 0.05. Tukey’s test was used to determine the significant difference in the data.

## 4. Conclusions

The combination of AEW at pH 2.5 and PL 1.5 J/cm^2^ was the most preferred treatment with the highest consumer acceptance. In terms of structural quality, the combination of AEW and PL 1.5 J/cm^2^ resulted in plant cells with the most intact structure. The treatment was able to retain the major bioactive compounds naturally present in the leaves of pennywort. In conclusion, the combined treatment retained the sensorial, structural and functional quality of the leaves up to 12 days of storage at 4 ± 1 °C. The influence of pulsed light in combination with acidic electrolysed water showed that throughout the storage period, the quality of the bioactive compound was higher compared to the control sample, resulting in the good acceptance by consumers on the sensorial properties as the morphological changes were slower for acidic EW and PL-treated leaves of pennywort. The bioactive phytocompounds from triterpene glycosides, namely, asiaticoside, madecassoside, asiatic acid and madecassic acid in pennywort were treated using a combined treatment (AEW and PL1.5) and preserved throughout the twelve days of storage.

## Figures and Tables

**Figure 1 molecules-28-00311-f001:**
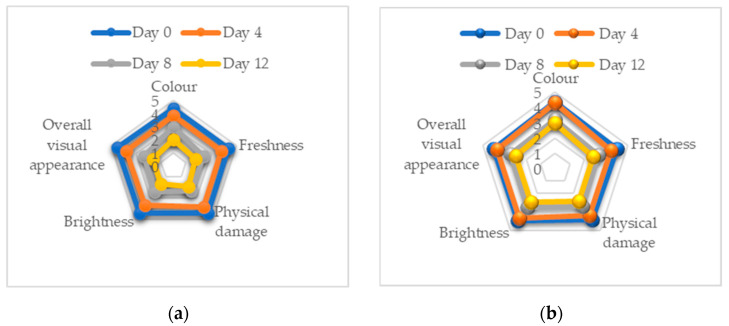
The spiderweb chart of sensory evaluation scores of fresh-cut pennywort leaves (**a**) untreated (tap water), (**b**) acidic electrolysed water, (**c**) PL + AEW treatment 1.5 J/cm^2^, (**d**) PL + AEW treatment 4.2 J/cm^2^, (**e**) mean values for the overall acceptance.

**Figure 2 molecules-28-00311-f002:**
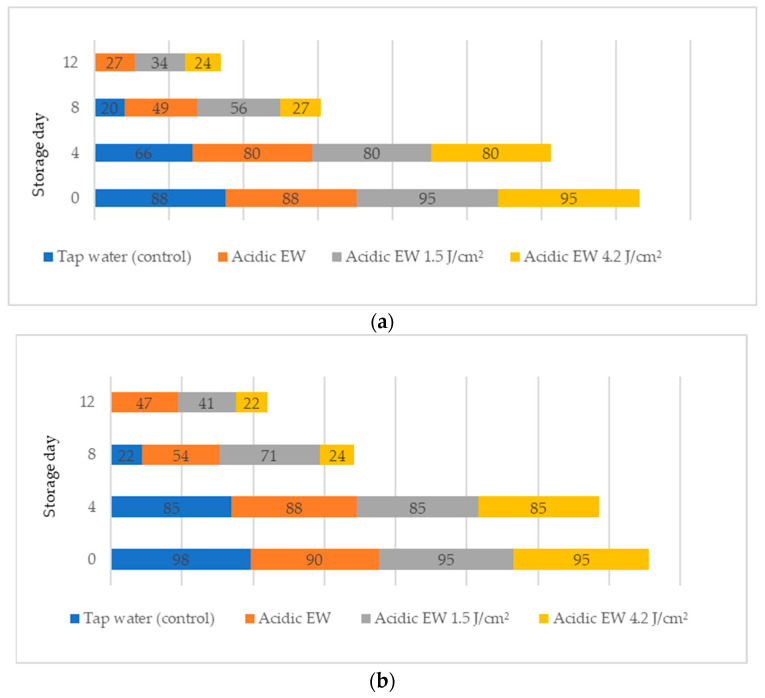
The sensory evaluation score on yes percentages (%) of fresh-cut pennywort leaves (**a**) question 1 and (**b**) question 2.

**Figure 3 molecules-28-00311-f003:**
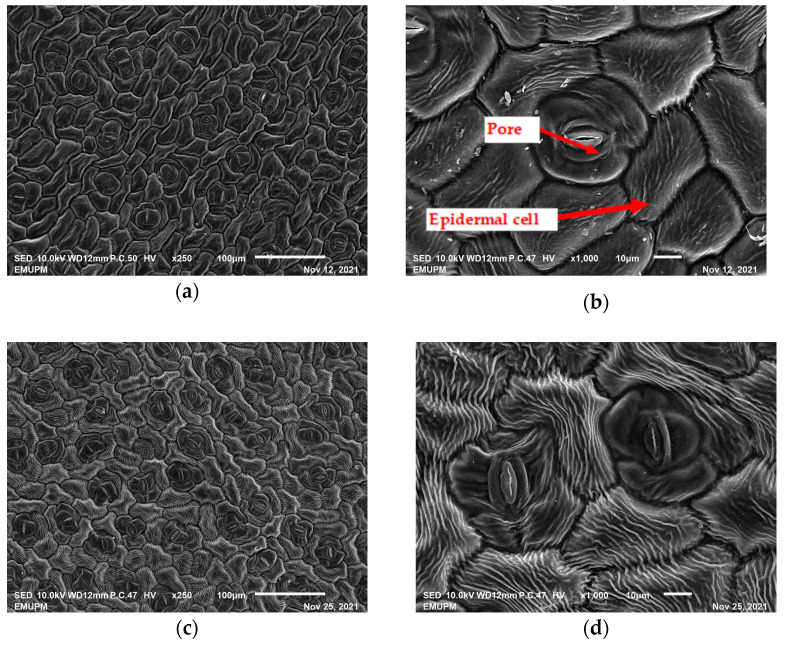
SEM micrographs showing surface structure of control pennywort leaves on the surface overview (**a**) and close up (**b**) on day 0, overview of the surface (**c**) and close up of a leaf (**d**) on day 12.

**Figure 4 molecules-28-00311-f004:**
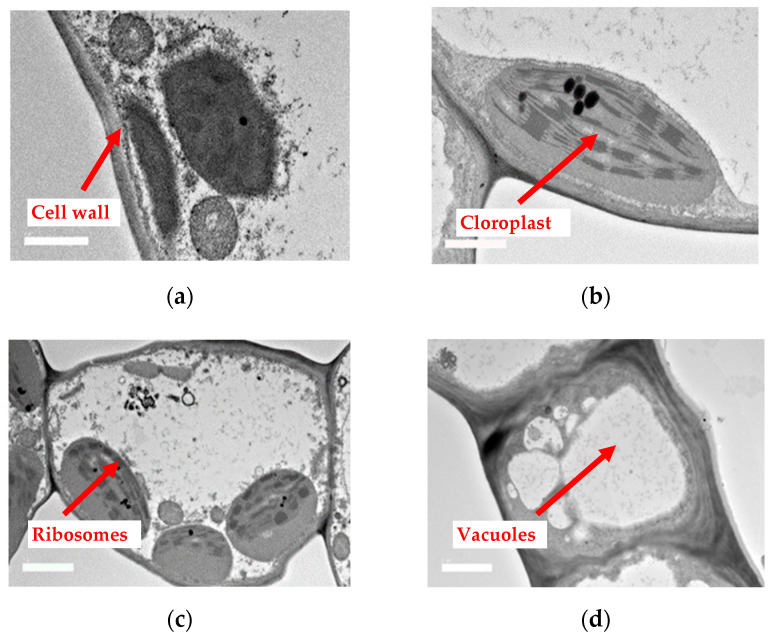
Transmission electron microscopy (TEM) analysis of leaf tissues from pennywort leaves (**a**) cell wall, (**b**) chloroplasts, (**c**) ribosomes and (**d**) vacuole.

**Figure 5 molecules-28-00311-f005:**
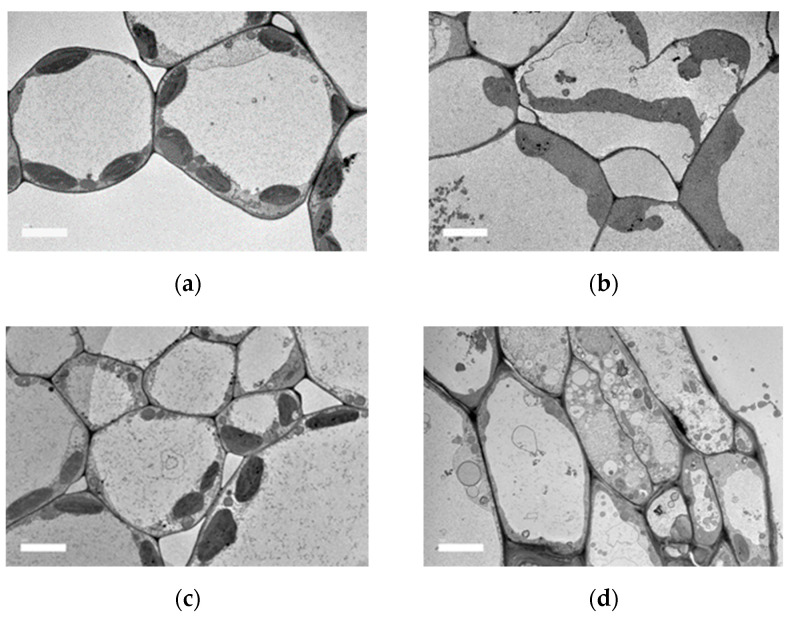
Transmission electron microscopy (TEM) analysis of leaf tissues from pennywort leaves control day 0 (**a**), day 12 (**b**) and AEW with PL 1.5 day 0 (**c**) and day 12 (**d**).

**Figure 6 molecules-28-00311-f006:**
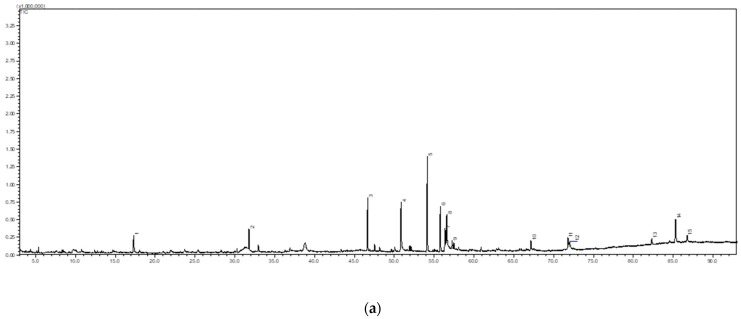
GC-MS chromatogram of fresh-cut pennywort leaves using ethanolic extract (**a**) control, (**b**) PL-treated sample.

**Table 1 molecules-28-00311-t001:** Scanning electron microscopic (SEM) images of fresh-cut pennywort leaves on day 0 and 12 of storage period.

Day/ Treatment	0	12
Acidic EW	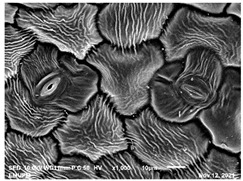	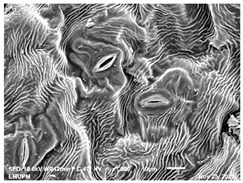
Acidic EW 1.5 J/cm^2^	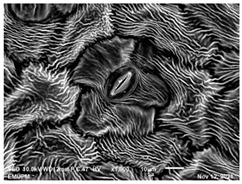	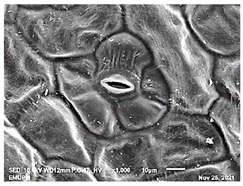
Acidic EW 4.2 J/cm^2^	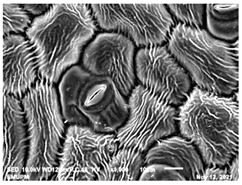	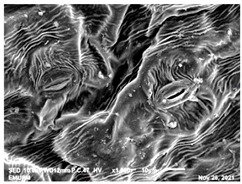
Acidic EW 6.9 J/cm^2^	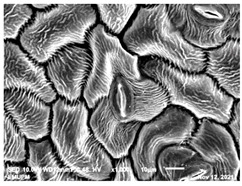	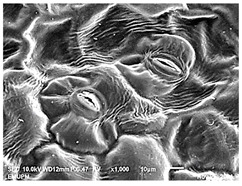

**Table 2 molecules-28-00311-t002:** Major constituents of fresh-cut pennywort leaves using ethanolic extract in GC-MS chromatogram (**a**) control, (**b**) PL-treated sample.

**(a) Peak** **No.**	**Name of Compound**	**Functional** **Compound**	**Molecular Formula**	**MW**	**High %**
1	2,3-Dihydro-3,5-dihydroxy-6-methyl-4h-pyran-4-one	Flavonoid	C_6_H_8_O_4_	144	3.93
2	Bergamotene <beta-, trans->	Sesquiterpenoids	C_15_H_24_	204	4.98
3	Neophytadiene	Diterpenes	C_20_H_38_	278	13.1
4	Hexadecanoic acid <n->	Fatty acids	C_16_H_32_O_2_	256	11.9
5	Panaxynone	Fatty acids	C_17_H_22_O	242	23.8
6	Phytol	Fatty acids	C_20_H_40_O	296	11.3
7	9,12-Octadecadienoic acid (Z,Z)-	Fatty acids	C_18_H_33_O_2_	280	5.31
8	Linolenate <methyl->	Fatty acids	C_19_H_32_O_2_	292	8.31
9	Ethyl-9,12-octadecadienoate	Fatty acids	C_20_H_36_O_2_	308	2.06
10	Hexadecanoic acid, 2-hydroxy-1-(hydroxymethyl)ethyl ester	Fatty acids	C_19_H_38_O_4_	330	2.32
11	9,12-Octadecadienoic acid (Z,Z)-, 2,3-dihydroxypropyl ester	Fatty acids	C_21_H_38_O_4_	354	2.65
12	Methyl (Z)-5,11,14,17-eicosatetraenoate	Fatty acids	C_21_H_34_O_2_	318	1.72
13	(+/−)-.alpha.-Tocopherol	Sterol	C_29_H_50_O_2_	430	1.37
14	Stigmasterol	Sterol	C_29_H_48_O	412	5.71
15	Rhamnol	Sterol	C_29_H_50_O	414	1.54
**(b) Peak** **No.**	**Name of Compound**	**Functional** **Compound**	**Molecular Formula**	**MW**	**High %**
1	2,3-Dihydro-3,5-dihydroxy-6-methyl-4h-pyran-4-one	Flavonoid	C_6_H_8_O_4_	144	4.67
2	Neophytadiene	Diterpenes	C_20_H_38_	278	1.84
3	7,11,15-Trimethyl-3-methylenehexadec-1-ene	Diterpenes	C_20_H_39_	278	1.84
4	Hexadecanoic acid <n->	Fatty acids	C_16_H_32_O_2_	256	12.5
5	Isopropyl palmitate	Fatty acids	C_19_H_38_O_2_	298	3.77
6	Panaxynone	Fatty acids	C_17_H_22_O	242	22.2
7	Phytol	Fatty acids	C_20_H_40_O	296	5.34
8	Linoelaidic acid	Fatty acids	C_18_H_32_O_2_	292	5.99
9	9,12,15-Octadecatrienoic acid, (Z,Z,Z)	Fatty acids	C_20_H_36_O_2_	308	11.1
10	Hexadecanoic acid, 2-hydroxy-1-(hydroxymethyl)ethyl ester	Fatty acids	C_19_H_38_O_4_	330	2.65
11	9,12-Octadecadienoic acid (Z,Z)-, 2,3-dihydroxypropyl ester	Fatty acids	C_21_H_38_O_4_	354	2.84
12	Linolenic acid, methyl ester	Fatty acids	C_19_H_32_O_2_	292	1.99
13	(+/−)-.alpha.-Tocopherol	Sterol	C_29_H_50_O_2_	430	1.78
14	Stigmasterol	Sterol	C_29_H_48_O	412	6.67
15	Rhamnol	Sterol	C_29_H_50_O	414	2.47

**Table 3 molecules-28-00311-t003:** Bioactive compound of madecassoside (MS), madecassic acid (MA), asiatic acid (AA) and asiaticoside (AS) in pennywort leaves.

Storage		Day 0		
Bioactive Compound/Treatment	MS	MA	AS	AA
Untreated (tap water)	11.65 ± 0.05 ^Aa^	1.16 ± 0.01 ^Db^	8.22 ± 0.27 ^Aa^	1.16 ± 0.01 ^Db^
Acidic EW	10.05 ± 0.05 ^ABa^	1.95 ± 0.01 ^Cb^	5.95 ± 0.07 ^Ba^	1.95 ± 0.01 ^Cb^
Acidic EW 1.5 J/cm^2^	7.63 ± 0.12 ^ABa^	2.69 ± 0.03 ^Bb^	5.75 ± 0.07 ^BCb^	2.69 ± 0.03 ^Bb^
Acidic EW 4.2 J/cm^2^	10.05 ± 0.05 ^ABa^	2.73 ± 0.09 ^Bb^	5.73 ± 0.38 ^BCa^	2.73 ± 0.09 ^Bb^
Acidic EW 6.9 J/cm^2^	6.41 ± 0.09 ^Ba^	2.94 ± 0.01 ^Ab^	5.05 ± 0.05 ^Ca^	2.94 ± 0.01 ^Ab^
**Storage**		**Day 12**		
Untreated (tap water)	3.52 ± 0.21 ^Cb^	2.63 ± 0.06 ^Ca^	1.79 ± 0.03 ^Cb^	2.63 ± 0.06 ^Ca^
Acidic EW	2.63 ± 0.02 ^Cb^	2.21 ± 0.01 ^Da^	2.38 ± 0.06 ^Cb^	2.21 ± 0.01 ^Da^
Acidic EW 1.5 J/cm^2^	7.22 ± 0.18 ^Aa^	3.46 ± 0.06 ^Ba^	6.27 ± 0.02 ^Aa^	3.46 ± 0.06 ^Ba^
Acidic EW 4.2 J/cm^2^	5.86 ± 0.13 ^ABa^	3.72 ± 0.07 ^ABa^	3.41 ± 0.08 ^Bb^	3.72 ± 0.07 ^ABa^
Acidic EW 6.9 J/cm^2^	4.13 ± 1.14 ^BCa^	3.87 ± 0.12 ^Aa^	2.36 ± 0.48 ^Cb^	3.87 ± 0.11 ^Aa^

Notes: Different uppercase letters indicate significant differences between treatment (*p* ≤ 0.05). Different lowercase letters indicate significant differences between storage day (*p* ≤ 0.05).

## Data Availability

The data presented in the present work are available upon reasonable request to the corresponding author.

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
