# Peer review of "Effect of Acidic Electrolysed Water and Pulsed Light Technology on the Sensory, Morphology and Bioactive Compounds of Pennywort (Centella asiatica L.) Leaves"

_molecules, 2022, doi:10.3390/molecules28010311_

Round 1

Reviewer 1 Report

The article ''Effect of acidic electrolysed water and pulsed light technology 2 on the sensory, morphology and bioactive compounds of pen- 3 nywort (Centella asiatica L.) leaves'' is presented for review. There are many minor and major recommendations before the article is considered for acceptance in this Journal. Following are the guided points for the authors to improve the article:

In the introduction part, the logicality is not very explicit, and there are some English grammatical errors, please double-check and correct them.

Authors are encouraged to cite some relevant and recent literature, such as.: Wu, W. Z., et al. "Expression and antibody preparation of Small Ubiquitin-like Modifier (SUMO) from Aspergillus flavus." IOP Conference Series: Earth and Environmental Science. Vol. 346. No. 1. IOP Publishing, 2019., Functional Analysis of Peptidyl-prolyl cis-trans Isomerase from Aspergillus flavus." International Journal of Molecular Sciences 20.9 (2019): 2206.  In order to strengthen their ideas in the light of previous work.

It would be appreciated if the authors can draw a table for a better comparison, in the main text.

Author Response

Thanks a lot for giving us the opportunity to submit the revised article to your progressive and valuable journal Molecules. We have answered (point by point) and revised the manuscript according to the reviewer’s comments. 

Comment 1: In the introduction part, the logicality is not very explicit, and there are some English grammatical errors, please double-check and correct them.

Answer: First we should be thankful to you for your valuable input that will help us to improve this article. We have gone through the entire manuscript carefully and after checking the introduction part, we have discussed and corrected it accordingly in the revised manuscript.

Comment 2: Authors are encouraged to cite some relevant and recent literature, such as.: Wu, W. Z., et al. "Expression and antibody preparation of Small Ubiquitin-like Modifier (SUMO) from Aspergillus flavus." IOP Conference Series: Earth and Environmental Science. Vol. 346. No. 1. IOP Publishing, 2019., Functional Analysis of Peptidyl-prolyl cis-trans Isomerase from Aspergillus flavus." International Journal of Molecular Sciences 20.9 (2019): 2206.  In order to strengthen their ideas in light of previous work.

Answer: Thank you for the comment. Based on the article mentioned by the reviewer, there was no relevant information suitable to be cited in this manuscript.

Comment 3: It would be appreciated if the authors can draw a table for a better comparison, in the main text.

Answer: We thanked the reviewer for the recommendation. However, we need more details on which part to add to the table as we already provide sufficient information in the manuscript.

Reviewer 2 Report

The authors of the manuscript titled " effect of acidic electrolyzed water and pulsed light technology on the sensory, morphology and bioactive compounds of pennywort leaves” attempted to determine the influence of pulsed light in combination with acidic electrolyzed water on the sensory, morphological changes and bioactive components in pennywort leaves during storage. The topic is quite interest, and it fits into the scope of Molecules. However, there are number of flaws in the experiment design and execution that prevents it from acceptance:

1.       Why in this manuscript M&M behind the result and discussion.

2.       Acidic electrolyse water or electrolyzed water? Keep consistant.

3.       Materials and Methods, the first should be materials, then treatments. How many grams or packed leaves for each treatment? For PLtreatment, each treatment packed in to a how large package? There are many information missed.

4.       Before sensory, we usually put the samples at room temperature for a while, did you do this? Or you directly go to do sensory?  Please describe detail.

5.       Figures need to improve, it’s difficult to understand. Also, it’s not clear, especially figure 1, 2 and 6.

Author Response

Thanks a lot for giving us the opportunity to submit the revised article in your progressive and valuable journal in the Molecules. We have answered (point by point) and revised the manuscript according to the reviewer’s comments.

Comment: The authors of the manuscript titled " Effect of acidic electrolyzed water and pulsed light technology on the sensory, morphology and bioactive compounds of pennywort leaves” attempted to determine the influence of pulsed light in combination with acidic electrolyzed water on the sensory, morphological changes and bioactive components in pennywort leaves during storage. The topic is quite interest, and it fits into the scope of Molecules. However, there are number of flaws in the experiment design and execution that prevents it from acceptance:

Why in this manuscript M&M behind the result and discussion.

Answer: First we should be thankful to you for your valuable input that will help us to improve this article. We use the template provided on the Molecules website (https://www.mdpi.com/journal/molecules/instructions) which is the arrangement of material and method behind the result and discussion.

Comment: Acidic electrolyse water or electrolyzed water? Keep consistent.

Answer: Thank you. The acidic electrolysed water was used in the manuscript and has been revised to the article.

Comment: Materials and Methods, the first should be materials, then treatments. How many grams or packed leaves for each treatment? For PL treatment, each treatment packed in to a how large package? There is many information missed.

Answer: Thank you for your valuable input. After checking, we revised the materials and method section with all the details. The data has been added and discussed accordingly in the revised manuscript.

Comment: Before sensory, we usually put the samples at room temperature for a while, did you do this? Or you directly go to do sensory?  Please describe detail.

Answer: Yes, the samples were at room temperature for a while for the short briefing before the evaluation started. The manuscript was revised accordingly to make the statement clearer.

Comment: Figures need to improve, it’s difficult to understand. Also, it’s not clear, especially figure 1, 2 and 6.

Answer:  The quality of the figures was improved.

Reviewer 3 Report

The work well organized and experimental designed set up, anyway some techniques used in this work seem to be showed the instrumental work not common use in the field of food processing to monitor the effect of process. Deeply in instrumental results, the author did not discuss deeply like any reader expected, such as SEM, TEM, or any spectroscopy methods useful for the discussion section. If the author can relate the results obtained from instrumental techniques with the discussion section would be more interesting to the reader. 

Author Response

We have answered (point by point) and revised the manuscript according to the reviewer’s comments. I hope it will satisfy the reviewers.

Comment: The work well organized and experimental designed set up, anyway some techniques used in this work seem to be showed the instrumental work not common use in the field of food processing to monitor the effect of process. Deeply in instrumental results, the author did not discuss deeply like any reader expected, such as SEM, TEM, or any spectroscopy methods useful for the discussion section. If the author can relate the results obtained from instrumental techniques with the discussion section would be more interesting to the reader

Answer: Thank you for your valuable input. We have gone through the entire manuscript carefully and revised the manuscript on the discussion part in subsections 3.2 and 3.3.

Round 2

Reviewer 1 Report

Revision is not satisfactory; many questions are remaining to provide suitable answers.

I recommend authors should do revision seriously. At this stage article has serious flaws, additional experiments are needed, and research is not conducted correctly.

Author Response

Manuscript Title: Effect of acidic electrolysed water and pulsed light technology on the
sensory, morphology, and bioactive compounds of pennywort (Centella asiatica
L.) leaves.

Reviewer 1.

Comment:  English very difficult to understand/incomprehensible

Answer: First we should be thankful to you for your valuable input that will help us to improve this article. We have gone through the entire manuscript carefully and sent our manuscript for proofreading services, we have discussed and corrected it accordingly in the revised manuscript.

Comment: Does the introduction provide sufficient background and include all relevant references?

Answer: Thank you for the comment. Based on the article mentioned by the reviewer, we added the article suggested in our manuscript and other references to provide sufficient background in the revised manuscript.

Comment: Is the research design appropriate? Are the methods adequately described?

Answer: Thank you for the comment. We explained the experimental design in the material and method part and revise the manuscript accordingly. (Refer to Materials and Methods part line 310)

Comment: Are the results clearly presented? Are the conclusions supported by the results?

Answer: We thanked the reviewer for the recommendation. We explained the result more clearer in the discussion part. The revised manuscript improved accordingly as we provide sufficient information in the manuscript.

Comment: Revision is not satisfactory; many questions are remaining to provide suitable answers.

I recommend authors should do revision seriously. At this stage article has serious flaws, additional experiments are needed, and research is not conducted correctly.

Answer: Thank you for the comment. The manuscript was improved accordingly. We have provided sufficient information in the manuscript.

Previous comment (round 1)

Comment: In the introduction part, the logicality is not very explicit, and there are some English grammatical errors, please double-check and correct them.

Answer: First we should be thankful to you for your valuable input that will help us to improve this article. We have gone through the entire manuscript carefully and sent our manuscript for proofreading services, we have discussed and corrected it accordingly in the revised manuscript.

Comment: Authors are encouraged to cite some relevant and recent literature, such as.: Wu, W. Z., et al. "Expression and antibody preparation of Small Ubiquitin-like Modifier (SUMO) from Aspergillus flavus." IOP Conference Series: Earth and Environmental Science. Vol. 346. No. 1. IOP Publishing, 2019., Functional Analysis of Peptidyl-prolyl cis-trans Isomerase from Aspergillus flavus." International Journal of Molecular Sciences 20.9 (2019): 2206.  In order to strengthen their ideas in the light of previous work.

Answer: Thank you for the comment. We rearranged the revised manuscript based on the article mentioned by the reviewer, we added the article suggested in our manuscript and other references to provide sufficient background in the revised manuscript. (Refer to line 79, references number 19)

Comment: It would be appreciated if the authors can draw a table for a better comparison, in the main text.

Answer: Thank you for the comment. For the comment above, we were not clear on what comparison the reviewer was referring to. Therefore, we did not include any table so as to not make the wrong assumptions. However, in experimental design, we explain the treatment (Refer 4.12 line 515).

Reviewer 2 Report

No

Author Response

Reviewer 2.

Comment: English language and style are fine/minor spell check required

Answer: First we should be thankful to you for your valuable input that will help us to improve this article. We have gone through the entire manuscript carefully and sent our manuscript to a professional proofreading service. We have discussed and made improvements accordingly in the revised manuscript.
